# Application of a Conducting Poly-Methionine/Gold Nanoparticles-Modified Sensor for the Electrochemical Detection of Paroxetine

**DOI:** 10.3390/polym13223981

**Published:** 2021-11-17

**Authors:** Saedah R. Al-Mhyawi, Riham K. Ahmed, Rasha M. El Nashar

**Affiliations:** 1Department of Chemistry, College of Science, University of Jeddah, Jeddah 22233, Saudi Arabia; sraszyad@gmail.com; 2Chemistry Department, Faculty of Science, Cairo University, Giza 12613, Egypt; rihamkamal93@gmail.com

**Keywords:** paroxetine, poly (dl-methionine), electropolymerization, selective serotonin reuptake inhibitor, voltammetry, gold nanoparticles

## Abstract

This work demonstrates a facile electropolymerization of a dl-methionine (dl-met) conducting polymeric film on a gold nanoparticle (AuNPs)-modified glassy carbon electrode (GCE). The resulting sensor was successfully applied for the sensitive detection of paroxetine·HCl (PRX), a selective serotonin (5-HT) reuptake inhibitor (SSRIs), in its pharmaceutical formulations. The sensor was characterized morphologically using scanning electron microscopy (SEM) with energy dispersive X-ray spectroscopy (EDX) and atomic force microscopy (AFM) and electrochemical techniques such as differential pulse voltammetry (DPV), electrochemical impedance spectroscopy (EIS) and cyclic voltammetry (CV). The proposed sensor, poly (dl-met)/AuNPs-GCE, exhibited a linear response range from 5 × 10^−11^ to 5 × 10^−8^ M and from 5 × 10^−8^ to 1 × 10^−4^ M using DPV with lowest limit of detection (LOD = 1 × 10^−11^ M) based on (S/N = 3). The poly (dl-met)/AuNPs-GCE sensor was successfully applied for PRX determination in three different pharmaceutical formulations with percent recoveries between 96.29% and 103.40% ± SD (±0.02 and ±0.58, respectively).

## 1. Introduction

Paroxetine hydrochloride hemihydrate (PRX, (3*S*,4*R*)-4-(4-fluorophenyl)-3-(3,4-methylenedioxyphenoxymethyl)piperidine) is known as a selective serotonin (5-HT) reuptake inhibitor (SSRIs) antidepressant drug, used to treat depression, panic disorders, generalized and social anxiety disorders, phobias, posttraumatic stress disorders and obsessive compulsive disorders [1]. Because of its tolerability, clinical effectiveness, and favorable side effect profile, after fluoxetine, PRX has become the second most often prescribed SSRI antidepressant [2,3].

This drug is believed to have therapeutic effects on the brain by working as a highly selective inhibitor of the serotonin transporter (SERT) and norepinephrine transporter (NET). SERT is a membrane protein that transfers the neurotransmitter serotonin from synaptic gaps to presynaptic neurons, where it is reabsorbed, and its function is terminated. PRX helps neurons, platelets, and other cells to store serotonin, a chemical neurotransmitter that impacts human emotions and motivation. Paroxetine blocks serotonin reuptake, raising serotonin levels in the presynaptic cleft and restoring normal function in diagnosed patients. On the other hand, PRX acts as a monoamine transporter that carries dopamine and the neurotransmitters norepinephrine (noradrenaline) from the synapse back to the cytosol, where they are combined with other transporters to form vesicles for subsequent storage and release. As a result, PRX inhibits the noradrenaline transporter from doing its function, allowing noradrenaline to persist in the synapse for longer rendering normal levels of noradrenaline to be reached in humans [2,4,5].

Several analytical methods have been reported in literature to quantify paroxetine such as high performance liquid chromatography [6,7,8,9], gas chromatography [10], capillary electrophoresis [1], spectrophotometry [7,10], and spectrofluorimetry [11,12]. Electrochemical analysis offers the benefits of simplicity, high sensitivity, and selectivity for the detection of analytes in various samples as compared to other analytical detection techniques [11,12,13].

A limited number of electrochemical methods have been reported for PRX detection. The most recent involves the use of a pencil graphite electrode. A gold disc electrode modified by cytochrome P450-2D6 enzyme encased in poly(8-anilino-1-napthalene sulphonic acid) was also developed for the detection of paroxetine and fluvoxamine [2]. In another work, PRX was detected by a boron-doped diamond electrode (BDDE) and an edge plane graphite electrode (EPGE) by means of square wave anodic stripping voltammetry (SWAdSV) [14]. Earlier, a Nafion- and MWCNTs-modified glassy carbon electrode was also developed and PRX was detected by differential pulse voltammetry (DPV) [1]. To the best of our knowledge, no reported method involves however modified GCE electropolymerized amino acids/gold nanoparticles.

Because of its simplicity of preparation, perfect control over the thickness of the membrane, broad variety of electrode materials, and easily controlled potential window, electropolymerization is a promising tool for the immobilization of conductive polymers on an electrode surface [15,16] due to the presence of amine and carboxylic groups in the amino acid structure, which facilitate their electropolymerization process [17]. Different amino acids were used for the production of a layer of conductive polymer with electrocatalytic activity, as well as the formation of additional active sites available for target interaction with analytes, enhancing the sensor’s sensitivity and selectivity [15,18].

Based on the literature, many amino acids were successfully electropolymerized. Very recent examples involve L-alanine electropolymerization in basic media using cyclic voltammetry (CV) for simultaneous detection of dopamine, ascorbic acid, serotonin and guanine [19] and poly(asparagine) electropolymerization in neutral medium on the surface of a carbon nanotubes and graphene mixed paste electrode for the electrochemical detection of acetaminophen. The electropolymerisation mechanism involves the removal of one hydrogen atom from the amino group of asparagine, a radical form of amine linked to the bare electrode [20]. For the detection of dopamine, aspartic acid (AS) and melamine (MEL) were electropolymerized on the surface of a carbon paste electrode in a binary mixture of AS:MEL (*v*/*v* = 1:1) [21] and poly(cysteine) was formed on pencil graphite electrode through CV electropolymerization in neutral media for the simultaneous determination of hydrazine and hydroxylamine [22]

dl-Methionine is one of the amino acids that can be easily electropolymerized onto an electrode surface as a porous conducting polymer using the CV technique in a potential window ranging from −0.8 V to 2 V at scan rate 0.1 V/s [17,23]. Many poly(methionine)-based sensors were reported in the literature for the sensing purposes and detection of different analytes such as food colorants [18], dopamine [15], mercury(II) ion [17] and dopamine and uric acid [24].

Gold nanoparticles (AuNPs) have been used to modify glassy carbon electrodes, due to their unique properties of high surface area, high conductivity, chemical stability, good biocompatibility and fast electron transfers ability [17,25,26], that make them highly used in modifying many electrochemical sensors, as reported in the detection of dopamine [27], enzymatic glucose [28], acetaminophen [29] and methyl mercury [30].

In this work, for the first time, a PRX sensor was developed based on the tendency of AuNPs to covalently bind to the surface of numerous polymers having functional groups such as –NH_2_, –CN and –SH [17,24]. Thus, electropolymerized dl-methionine (poly (dl-met) which has high potential for interactions with AuNPs is formed, resulting in improvement of the sensitivity and performance of the modified glassy carbon electrode (GCE) towards PRX in its pure form and in pharmaceutical formulations.

## 2. Experimental

### 2.1. Reagents and Apparatus

All reagents used in this work were of analytical grade, used without any further purification. Paroxetine hydrochloride hemihydrate (PRX) was obtained from the National Organization for Drug Control and Research (NODCAR, Giza, Egypt). Paroxetine CR^®^ 15, 25, and 37.5 mg/tab were purchased from the local market. dl-Methionine ≥ 99%, gold (III) chloride trihydrate 99.9%, D-glucose ≥ 99.5%, lactose, ascorbic acid 99%, urea, potassium dihydrogen phosphate anhydrous, dipotassium hydrogen orthophosphate, ethanol 99%, sulfuric acid 98%, hydrochloric acid 36%, potassium ferricyanide, potassium ferrocyanide trihydrate and potassium chloride were purchased from Sigma‒Aldrich (Hamburg, Germany). All solutions were prepared in Milli-Q water purified in a Purelab UHQ system (ELGA, High Wycombe, UK).

All electrochemical measurements were performed using a DY2113 mini potentiostat (Digi-Ivy, Inc., Austin, TX, USA). EIS measurements were carried out using a Palmsens4 EIS potentiostat/galvanostat (Palmsens BV, Houten, The Netherlands) using a three-electrode cell, consisting of a reference electrode [CHI‒150 Saturated Calomel Electrode (SCE)], a working electrode [CHI-104 Glassy carbon electrode (GCE) 3 mm in diameter] (CH Instruments Inc., Bee Cave, TX, USA), and 1 mm platinum wire as an auxiliary electrode. pH measurements were performed using a digital Jenway pH meter, model 3510 (Jenway Instruments, Staffordshire, UK). Surface topographic characterization was performed using a 5600 Ls atomic force microscopy (AFM) system, (Agilent Technologies, Inc., Santa Clara, CA, USA) in the contact mode, and field emission electron microscopy (FESEM, Sigma 300VP, Zeiss, Jena, Germany).

### 2.2. Preparation of Solutions

Stock solution of 1 mM of paroxetine was prepared in ethanol, and then diluted in PBs pH 7.5 to the required concentrations. Phosphate buffer solution (PBs) was prepared by dissolving appropriate amounts of potassium dihydrogen phosphate anhydrous and dipotassium hydrogen orthophosphate in ultra-pure water to yield 0.1 M PBs pH 7.5.

For sampling of pharmaceutical formulations, ten tablets of each formulation were ground in a mortar and weighed, then equivalent weights of one tablet were dissolved in 0.1 M HCl: ethanol mixture (1:5), afterwards the solutions were centrifuged at 5000 rpm for 10 min, filtered using (ultra-pure) PTFE 0.45 μm and diluted in 0.1 M PBs pH 7.5 to prepare the required concentrations.

### 2.3. Electrode Modification

Prior to modification, the surface of GCE was polished with alumina slurry (0.05 μm), rinsed with doubly distilled water, sonicated in an ultrasonic bath with ultra-pure water for 5 min and scanned in 0.1 M H_2_SO_4_ from 0 V to 1.5 V at a scan rate of 0.1 V/s until a stable signal was obtained. The cleaned GCE was then immersed in 0.5 M H_2_SO_4_ solution containing 1 mg/mL of HAuCl_4_ at constant potential of −0.2 V for different deposition times using chronoamperommetry (i-t) as previously reported for AuNPs deposition [27,29], then washed with distilled water and dried at room temperature. Afterwards, a conductive layer of poly (dl-met) was then fabricated on the surface of the modified electrode (AuNPs-GCE) using 0.5 mM of dl-methionine in 0.1 M phosphate buffer solution (PBs) pH 7 electropolymerized from −0.8 V to 1.5 V at scan rate 0.1 V/s for different number of cycles, then the electrode was thoroughly rinsed with distilled water to remove any unreacted methionine and kept dry till used.

### 2.4. Experimental Measurements of PRX

The electrochemical measurements were performed on the surface of poly (dl-met)/AuNPs-GCE employing DPV from 0.5 V to 1 V for a 10 µM paroxetine solution in PBs pH 7.5 at step potential 0.005 V, pulse period 0.2 s, pulse amplitude 0.1 V and pulse width 0.05 V. Electrochemical impedance spectroscopic measurements (EIS) were recorded at 0.2 V and frequency range (0.1 Hz–100 KHz) with amplitude 0.01 V using 1 mM [Fe(CN)_6_]^3/4−^ prepared in 0.1 M KCl as a supporting electrolyte. All measurements were performed at room temperature.

## 3. Results and Discussion

### 3.1. Optimization of the Sensor’s Fabrication Conditions

To achieve the optimum sensor’s performance, different experimental parameters affecting the electrode composite (deposition time of AuNPs, monomer concentration, and the number of electropolymerization cycles), pH of the analyte detection media, and electrochemical parameters (step potential, pulse amplitude, accumulation time, and accumulation potential) should be investigated.

#### 3.1.1. Effect of Deposition Time of AuNPs

Before electropolymerization of dl-met, the surface of GCE was modified at different deposition time intervals with AuNPs, and the *I*_PRX_ of 10 µM PRX on the AuNPs-GCE were recorded and compared with the response of the bare GCE. Figure 1A showed that the maximum *I*_PRX_ was at 80 s deposition of AuNPs-GCE. AuNPs increases the *I*_PRX_ by ~10-fold of the bare GCE, this may be due to AuNPs enhanced the electron transfer and increased surface area of the GCE thus increasing its sensitivity [25]. Appendix A presents the i-t curves of AuNPs at constant potential of −0.2 V for 80 s at which Au (III) is reduced to AuNPs and deposited on GCE surface [30].

#### 3.1.2. Monomer (dl-met) Concentrations

The effect of monomer concentration in the polymerization mixture on the oxidation peak of PRX was tested as it may affect the thickness of the polymeric membrane formed, and in turn, the conductivity of the surface. Different monomer concentrations ranging from 0.1 to 3.0 mM in PBs pH 7 were tested, while other variables such as potential window of (−0.3 to 1.5 V) at scan rate of 0.1 V/s, Au deposition time (80 s), five polymerization scan cycles and pH media of PRX (7.5) were kept constant, followed by recording the oxidation current (*I*_PRX_) of 10 µM PRX at pH 7.5. As shown in Figure 1B, 0.5 mM of dl-met was found to result in the highest peak current compared to other concentrations.

#### 3.1.3. Electropolymerization of dl-met

dl-Met (0.5 mM in 0.1 M PBs pH 7) was electropolymerized on a AuNPs-modified GCE using cyclic voltammetry (CV) in a potential range from −0.8 V to 1.5 V at scan rate 0.1 V/s at different number of polymerization cycles (3–12 cycles) cycles. From the CVs of electropolymerization shown in Figure 1C, an irreversible oxidation peak was observed at ~1 V in the first scan, which decreased and slightly shifted to more negative potential in the following scans indicating the formation of a polymeric layer of poly (dl-met) on the surface of AuNPs-GCE [17]. The thickness of the polymeric layer can be controlled by varying the number deposition cycles as it may affect the sensor’s performance. Low number of cycles may produce a thin unstable layer that can be easily leached from the sensors surface while high number of cycles may result in the formation of a compact polymeric insulating layer on the surface [31]. Based on the results shown in Figure 1D, it can be revealed that five polymerization cycles were sufficient to yield the maximum oxidation current of PRX, further increase led to a decrease in the current signal

### 3.2. Effect of pH

The pH of the medium plays an important role on the electrochemical activity of PRX. As previously reported, the highest oxidation peak current was found from Ph ~5.0 to ~9.0 [1,13] depending on the pK_a_ value of PRX (pKa = 9.8) [9]. In this work the influence of pH on the *I*_PRX_ of 10 µM PRX on the surface of poly (dl-met)/AuNPs-GCE was investigated in PBs pH (5.5–8.0). As shown in Figure 1E, the increase in pH resulted in an increase in *I*_PRX_ up to pH 7.5, which was chosen as the optimum pH value after which the *I*_PRX_ was noticed to decrease hardly, as previously reported by Piech et. al. that *I*_PRX_ decreased in more basic or acidic media [1]. Also, the increase in pH was found to shift the PRX peak potential to more negative values as seen in Figure 1F, indicating the irreversible oxidation process behavior [32]. Equation (1) indicated that the peak potential has a linear pH dependence for poly (dl-met)/AuNPs-GCE, as previously reported [14]:*E_p_* (*V*) = 1.385 − 0.079 pH; r = 0.996, (pH 5.5–8)(1)

The relation between this plot and number of electron and proton involved in the electrochemical process (Equation (2)), was clarified by Rieger:*E_p_* = k − (0.059 y/n) pH(2)
where n, y and k are the number of electrons, number of protons and the intercept of the linear relation [33]. Since the obtained slope value of 0.079 is very close to the ideal slope of 0.059, this indicates that equal numbers of protons and electrons are involved in the oxidation of PRX [33,34]. Based on the structure of PRX, the electrooxidation mechanism of PRX might involve the alkoxybenzene which can be transformed to the quinone form, in a similar approach to that reported for mebeverine and tamsulosin hydrochloride [34,35].

### 3.3. Effect of DPV Operational Parameters

To maximize the experimental performance of poly (dl-met)/AuNPs-GCE to detect PRX, different operational variables were examined such as potential step (E_s_), pulse amplitude (ΔE), accumulation time (t_.acc_) and accumulation potential (pot_.acc_), such variables are considered to be the most important parameters of DPV technique.

E_s_ was varied from 0.005 V to 0.035 V, while keeping other variables constant and *I*_PRX_ of 10 µM PRX at pH 7.5 was recorded, for 0.005 V the *I*_PRX_ was 13.5 µA after which the *I*_PRX_ was decreased as shown in Appendix A and for 0.030 V and 0.035 V the *I*_PRX_ were 10.48 and 10.70 µA, respectively. ΔE was varied from 0.05 V to 0.25 V, the results revealed that 0.1 V was the optimum applied pulse amplitude resulting in *I*_PRX_ of 14.7 µA after which peak deformation was noticed.

The accumulation time t_.acc_ and potential pot_.acc_ were also studied from 2–120 s and 0–0.05 V, respectively. Appendix A shows a decrease in the *I*_PRX_ on increasing the accumulation time up to 20 s, no improvement in response was noticed even on reaching accumulation times of 120 s thus, 2 s was chosen as the optimum accumulation time. Appendix A shows that the maximum *I*_PRX_ was achieved in case of applying zero pot_.acc_ was applied, further increase in pot_.acc_ starting from 0.01 V led to a decrease *I*_PRX._, thus, no pot_.acc_ was applied.

Figure 2 represents the DP voltammograms of 10 µM PRX on unmodified GCE (curve a) which showed a weak oxidation peak at ~0.8 V, the oxidation peak of PRX increased on AuNPs-GCE (curve b) due to the electroactivity and catalytic properties of AuNPs [36], poly (dl-met)/AuNPs-GCE showed the highest oxidation peak (curve c) as a result of the combination of AuNPs and conducting layer of poly (dl-met), a slight difference in *I*_PRX_ was shown on using poly (dl-met)-GCE (curve d) compared to bare GCE.

### 3.4. Surface Topography of the Sensor

Atomic force microscopy (AFM) was used to investigate the surface morphology and root mean square (RMS) roughness. The roughness was obtained from AFM image using contact mode and 2.5 µm × 2.5 µm scan area for GCE, AuNPs-GCE and poly (dl-met)/AuNPs-GCE. Figure 3A(I) showed a smooth surface of bare GCE with a RMS roughness value of 0.052 nm, while RMS roughness value of AuNPs-GCE, Figure 3A(II), increased to 10.9 nm indicating the formation of AuNPs, and while Figure 3A(III) for poly (dl-met)/AuNPs-GCE indicated that RMS roughness decreased to 0.124 nm due to formation of smooth layer of conducting polymer on AuNPs surface. This reduction in roughness will most likely result in a more compact and stable layer on the working electrode [17].

Scanning electron microscopy (SEM) and energy dispersive X-ray spectroscopy (EDX) were also used for surface characterization of GCE, AuNPs-GCE and poly (dl-met)/AuNPs-GCE, samples were fixed 5.00 mm working distance, at an excitation voltage of 5 KV and magnification of 10 KX. Figure 3B(I) shows a smooth surface of bare GCE, Figure 3B(II) presents the surface of the AuNPs-GCE displaying particle sizes ranging between 5.00 nm and 7.21 nm, indicating an increase in the electrode’s surface area. Particle size was measured through the Image J software (version 2.35 for Windows, 64 bit, National Institutes of Health, Bethesda, MD, USA). Figure 3B(III) presents the surface of poly (dl-met)/AuNPs-GCE which is smoother than that of AuNPs-GCE confirming the formation of dl-met polymeric layer.

EDX analysis was performed to investigate the elements presented on the surface of the modified sensors. The atomic percent of Au was found to be 0.00% for bare GCE and increased to 1.06% for AuNPs-GCE Confirming the electrodeposition of AuNPs on CGE and then decreased to 0.33% for poly (dl-met)/AuNPs-GCE, confirming the formation of a polymeric film on the electrode’s surface. Appendix A presents the corresponding EDX spectra of AuNPs-GCE (I) and poly (dl-met)/AuNPs-GCE (II).

### 3.5. Electrochemical Characterization of the Modified Sensor

The stepwise modification of the sensor was investigated electrochemically using cyclic voltammetry (CV) and electrochemical impedance spectroscopy (EIS) in 1 mM [Fe(CN)_6_]^3−/4−^ prepared in 0.1 M KCl. Figure 4A presents cyclic voltammograms of the reversible redox peak of [Fe(CN)_6_]^3−/4−^ on the bare GCE (curve a), which showed a weak redox peak compared to the AuNPs-modified GCE (curve b). The observed increase in the redox peak can be attributed to the electroactivity of AuNPs. A further increase in the redox peak was noticed on the surface of poly (dl-met)/AuNPs-GCE (curve c) which gave the highest redox peak, indicating the formation of a layer of a conducting polymer.

Comparison of the redox peak of poly (dl-met) directly deposited on the bare GCE (curve d), shows lower redox peaks compared to those obtained on depositing poly (dl-met) on the surface of AuNPs-GCE, indicating that there is a synergistic effect of AuNPs on enhancing the sensitivity and conductivity of the poly (dl-met) due to the interaction of (-SH) group of methionine with AuNPs [17] besides increasing the effective surface area of the GCE [25,29]. this agreed with previously reported sensors modified with methionine and AuNPs [17,23].

Electrochemical impedance spectroscopy (EIS) was also performed for each modified surface in 1 mM [Fe(CN)_6_]^3−/4−^ using PSTrace software. The charge transfer resistance (R_ct_) was obtained by fitting with Randle’s equivalent circuit [18]. Figure 4B shows Nyquist plots for the bare GCE (curve a), which exhibited the largest semicircle (R_ct_ = 330.5 Ω) due to the poor conductivity of GCE in comparison with AuNPs-GCE (curve b) where the semicircle decreased indicating the electroactivity of AuNPs film (R_ct_ = 101.4 Ω) and poly (dl-met)/AuNPs-GCE (curve c), which exhibited the smallest semicircle (R_ct_ = 69.1 Ω). poly (dl-met)-GCE (curve d) showed large semicircle (R_ct_ = 301.1 Ω), which didn’t exhibit a significant difference from bare GCE, confirming that the combination of AuNPs and poly (dl-met) layers provides a good conductivity for the modified GCE surface.

The modified sensor was also characterized using CV at different scan rates from 0.01 to 0.5 V/s, the obtained linear relationship between cathodic and anodic peak currents of [Fe(CN)_6_]^3−/4−^ and the square roots of the scan rate shown in Figure 5, confirm that the redox process of the GCE, AuNPs-GCE and poly (dl-met)/AuNPs-GCE was under diffusion control, respectively [37,38].

The working surface area of the sensor was calculated from the Randles-Sevcik Equation (Equation (3)):*I*_pa_ = (2.69 × 10^5^) n^3/2^ A (D_o_)^1/2^ C_o_ *v*^1/2^(3)
where n is the transferred electrons number (n = 1), *I*_pa_ is the anodic peak current, D_o_ is the diffusion coefficient (7.6 × 10^−6^ cm^2^/s), C_o_ is the concentration of the [Fe(CN)_6_]^3−/4−^ (1 × 10^−6^ mol/cm^3^) and A is the surface area of the electrode [38,39]. The surface area of the bare GCE, AuNPs-GCE and poly (dl-met)/AuNPs-GCE were calculated to be 0.0081, 0.0100 and 0.012 cm^2^, respectively, confirming the enhancement of conductivity of poly (dl-met)/AuNPs-GCE due to the increase of the active sensor surface.

### 3.6. Effect of Possible Interferents on the Response (Selectivity)

Certain compounds that are frequently found as components of tablets or biological media and various species with potential interference capabilities in the PRX measurement such as dopamine, urea, ascorbic acid, glucose, and lactose were tested. The interference effect of these compounds was examined by determining the *I*_PRX_ of 10 μM PRX in a solution containing equimolar, 10-fold, and 100-fold of these compounds under optimum sensor conditions. Dopamine, urea, ascorbic acid, glucose and lactose in their equimolar ratio with PRX were found to positively affect the response by 2.06%, 2.02%, 0.80%, 5.00% and 2.80%, in 10-fold ratio 0.62%, 1.32%, 4.82%, 2.72% and 1.47% and in 100-fold ratio 1.50%, 2.20%, 0.90%, 2.47% and 4.7%, respectively, which revealed that poly (dl-met)/AuNPs-GCE did not show any significant interference due to these interferents within the studied concentration ranges.

### 3.7. Validation of the Poly (dl-Met)/AuNPs-GCE Sensor Response

The standard calibration curve was constructed by measuring different concentrations of PRX, ranging from 5 × 10^−11^ to 1 × 10^−4^ M in 0.1 M PBs pH 7.5 under the optimized experimental conditions. Figure 6 shows the response oxidation peaks (DP voltammograms) of PRX on the surface of poly (dl-met)/AuNPs-GCE from 5 × 10^−11^ to 1 × 10^−4^ M, the peak current was found to increase with increasing PRX concentration. Two linear ranges were noticed the first a linear range, shown in Figure 6A was observed from 5 × 10^−11^ to 5 × 10^−9^ M, R^2^ = 0.9912, represented by regression Equation (4) and the second range from 5 × 10^−9^ to 1 × 10^−4^ M, R^2^ = 0.9950 represented by regression Equation (5) as shown in in Figure 6B. The limit of detection (LOD) obtained based on (S/N = 3) was found to be 1 × 10^−11^ M which indicated the sensitivity and lower LOD of the presented sensor compared to other previously reported methods given in Table 1:*I*_PRX_ = 14.959 + 1.234 log [PRX](4)
*I*_PRX_ = 26.889 + 2.635 log [PRX](5)

The repeatability of the poly (dl-met)/AuNPs-GCE sensor was investigated by recording *I*_PRX_ in 10 µM PRX at pH 7.5 five times within the same day on the same sensor prepared under the optimized conditions. The results revealed a good repeatability expressed as RSD% of n = 5 was 3.47%. The reproducibility of the poly (dl-met)/AuNPs-GCE was also tested by preparing three different sensors on three different days upon the same optimum conditions. The results showed that RSD% of n = 3 were in range of 3.37%–4.17% revealing the high reproducibility of the method. The stability of the sensor was tested by measuring 10 µM PRX at pH 7.5 at different time intervals over one month and the sensor was found to have a very stable response and retained up to 97% of its response.

### 3.8. Applications of the Designed Sensor for the Detection of Paroxetine in Pharmaceutical Formulations

The accuracy and applicability of poly (dl-met)/AuNPs-GCE sensor for PRX was determined in three different pharmaceutical formulations using the standard additions method at different concentrations [31]. Table 2 shows the recovery percentages for Paroxetine CR^®^ 12.5 mg/tab are varying from (99.10% to 10.3.40%; SD ± 0.21 to ± 0.58), Paroxetine CR^®^ 25 mg/tab (96.29% to 105.05%; SD ± 0.02 to ± 0.40) and Paroxetine CR^®^ 37.5 mg/tab (97.57% to 103.04%; SD ± 0.04 to ± 0.37) indicating that the developed poly (dl-met)/AuNPs-GCE sensor is suitable for PRX detection in its commercial dosage forms without interferences from any pharmaceutical additives or excipients.

## 4. Conclusions

A new electrochemical sensor for selective and sensitive detection of PRX was developed. The sensor was fabricated using a GCE modified with electrodeposited AuNPs at constant potential of −0.2 V for 80 s and electropolymerized dl-met from −0.8 V to 1.5 V for five cycles at a scan rate of 0.1 V s^−1^. The poly (dl-met)/AuNPs-GCE sensor showed a better conductivity and high surface area to detect 10 µM PRX at pH 7.5, step potential of 0.005 V, pulse amplitude of 0.1 V over bare GCE and AuNPs-GCE. The sensor showed high selectivity towards PRX in the presence of some common biological interferents such as glucose, lactose, urea, dopamine, and ascorbic acid. poly (dl-Met)/AuNPs-GCE showed linearity from 5 × 10^−11^ to 5 × 10^−8^ M and 5 × 10^−8^ to 5 × 10^−4^ M with detection limit of 1 × 10^−11^ M (S/N = 3) using DPV. The proposed sensor has the lowest LOD, good reproducibility and applicability in pharmaceutical detection of PRX compared to other previously reported electrochemical sensors, which represents a simple and cost-effective approach for PRX detection in regulator and quality control units in pharmaceutical companies.

## Figures and Tables

**Figure 1 polymers-13-03981-f001:**
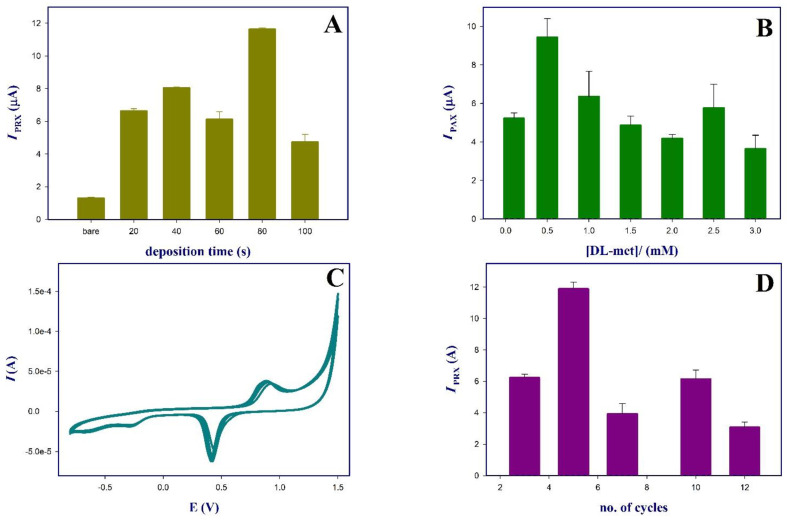
(**A**) Effect of AuNPs deposition time. (**B**) Effect of dl-met concentration on IPRX. (**C**) CV of five cycles electropolymerization of 0.5 mM dl-methionine in PBs pH 7 from −0.8 V to 1.5 V at scan rate 0.1 V s^−1^ on AuNPs-GCE. (**D**) Effect of number of scan cycles. Effect of pH media on the IPRX (**E**) and peak potential (**F**).

**Figure 2 polymers-13-03981-f002:**
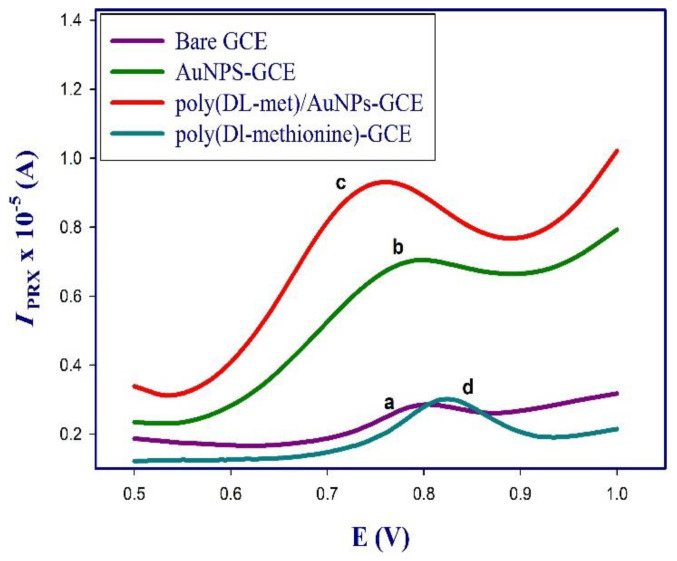
DPV of 10 µM PRX in PBs pH 7.5 on the surface of bare GCE (a), AuNPs-GCE (b), poly (dl-met)/AuNPs-GCE (c) and poly (dl-met)-GCE (d) from 0.5 to 1.0 V at step potential of 0.005 V and pulse amplitude of 0.1 V.

**Figure 3 polymers-13-03981-f003:**
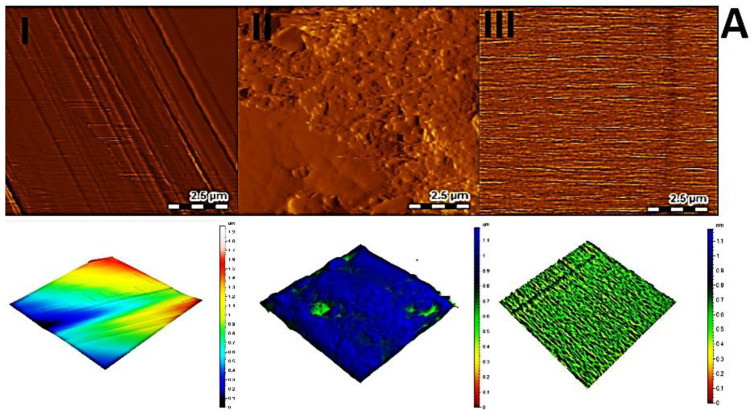
(**A**) AFM image of bare GCE (**I**), AuNPs-GCE (**II**) and poly (dl-met)/AuNPs-GCE (**III**). (**B**) SEM images of bare GCE (**I**), AuNPs-GCE (**II**) and poly (dl-met)/AuNPs-GCE (**III**).

**Figure 4 polymers-13-03981-f004:**
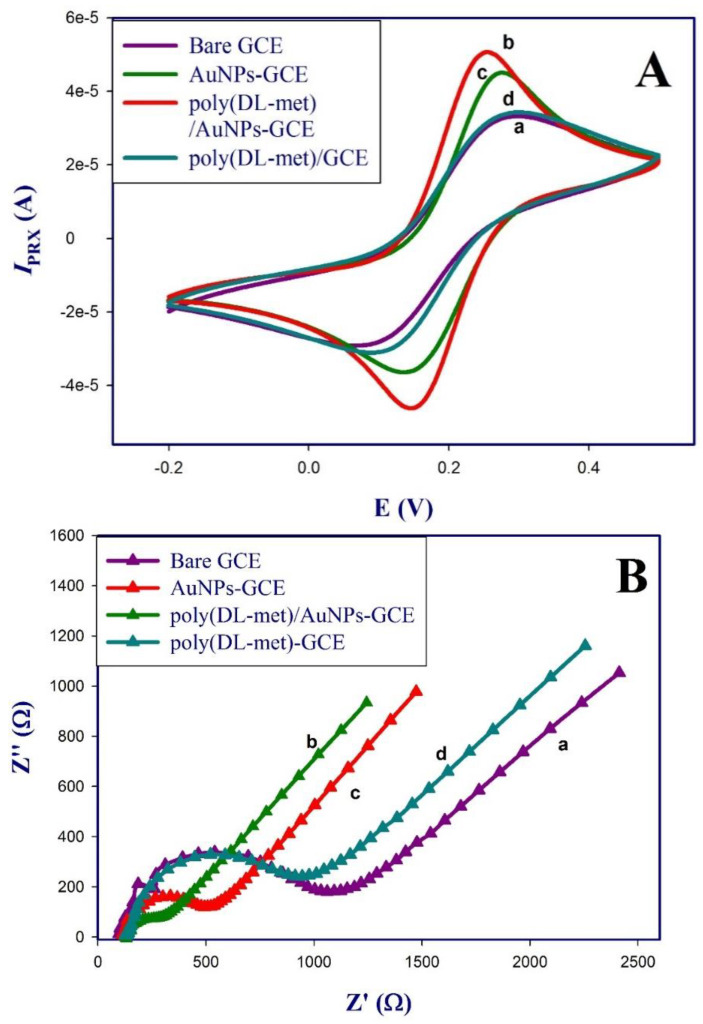
(**A**) Cyclic voltammograms of stepwise fabrication of poly (dl-met)/AuNPs-GCE from −0.2 V to 0.5 V at scan rate 0.05 V s^−1^ vs. SCE. Bare GCE (a), AuNPs-GCE (b), poly (dl-met)/AuNPs-GCE (c) and poly (dl-met)-GCE (d). (**B**) Nyquist plots of the stepwise fabrication of poly (dl-met)-AuNPs/GCE at potential of 0.2 V, frequency range (0.1 Hz–100 KHz) and amplitude 0.01 V.

**Figure 5 polymers-13-03981-f005:**
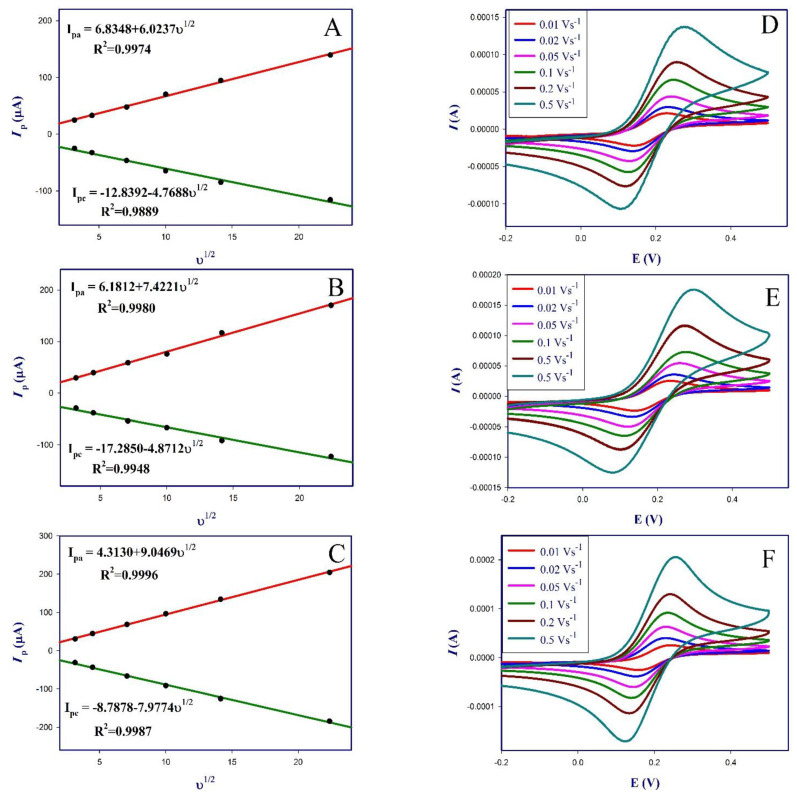
CV at different scan rates from 0.01 V s^−1^ to 0.5 V s^−1^ using 1 mM [Fe(CN)_6_]^3−/4−^ in 0.1 M KCl solution for: (**A**) Bare GCE, (**B**) AuNPs-GCE and (**C**) poly (dl-met)/AuNPs-GCE. Linear relationship between the anodic and cathodic peak currents of [Fe(CN)_6_]^3−/4−^ and square root of the scan rate for: (**D**) Bare GCE, (**E**) AuNPs-GCE and (**F**) poly (dl-met)/AuNPs-GCE.

**Figure 6 polymers-13-03981-f006:**
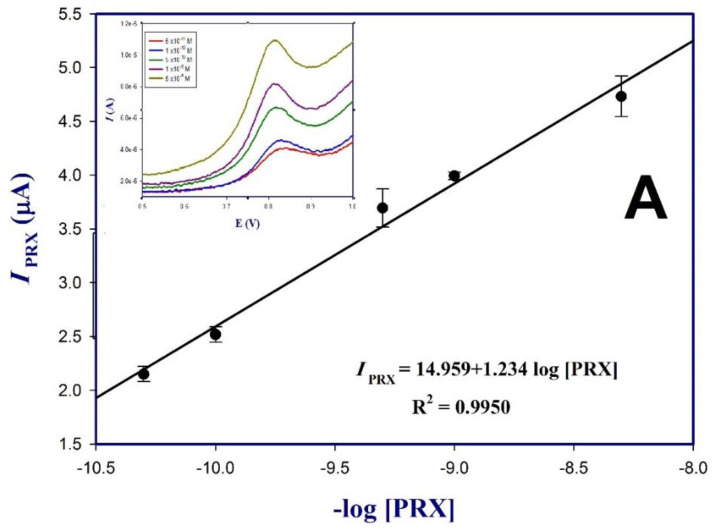
Calibration graph for PRX on the surface of poly (dl-met)/AuNPs-GCE for concentrations (**A**) 5 × 10^−11^ to 1 × 10^−4^ M and (**B**) 5 × 10^−8^ to 1 × 10^−4^ M at optimum conditions (inset: DPV of the tested concentrations).

**Table 1 polymers-13-03981-t001:** Comparison between the present work and other previously reported electrochemical methods for PRX detection.

Sensor Material	Analytical Method	Linear Range (M)	LOD (M)	R^2^	Reference
poly (dl-met)/AuNPs-GCE	DPV	5 × 10^−11^–5 × 10^−9^	1.00 × 10^−11^	0.991	Present work
		5 × 10^−9^–1 × 10^−4^		0.995	
rGO/PWA/PGE	DPV	9 × 10^−8^–1 × 10^−6^	9.00 × 10^−10^	0.998	[13]
Au/PANSA/CYP2D6	DPV	0.5 × 10^−8^–0.5 × 10^−7^	2.00 × 10^−9^	0.990	[2]
BDDE	SWAdSV	7.0 × 10^−7^–3.5 × 10^−6^	6.95 × 10^−9^	0.999	[14]
EPGE	SWAdSV	1.0 × 10^−8^–5.0 × 10^−6^	1.03 × 10^−9^	0.999	[14]
Nafion-MWCNTs/GCE	DPV	0.1 × 10^−6^–2.5 × 10^−6^	0.62 × 10^−7^	0.998	[1]

PWA: Phosphotungstic acid. PGE: Pencil graphite electrode. PANSA: poly (8-anilino-1-napthalene sulphonic acid). CYP2D6: cytochrome P450-2D6 enzyme. BDDE: Boron-doped diamond electrode. EPGE: Edged pencil graphite electrode. SWAdSV: square wave anodic stripping voltammetry.

**Table 2 polymers-13-03981-t002:** Application of the developed sensor for the determination of PRX in pharmaceutical formulations.

Sample	Taken (µg mL^−1^)	Found (ng mL^−1^)	Recovery% ± SD	RSD%
Paroxetine CR^®^ 37.5	0.27	0.28	103.04% ± 0.05	0.46
mg/tab	0.99	0.97	97.57% ± 0.37	3.11
	2.87	2.91	98.62% ± 0.04	2.98
Paroxetine CR^®^ 25	0.27	0.26	96.29% ± 0.07	1.24
mg/tab	0.99	1.04	105.05% ± 0.02	0.21
	2.87	2.917	101.16% ± 0.40	2.98
Paroxetine CR^®^ 12.5	0.27	0.27	103.40% ± 0.58	5.41
mg/tab	0.99	0.98	99.10% ± 0.21	1.80
	2.87	2.94	102.50% ± 0.41	3.05

SD: Standard deviation (n = 3); RSD: Relative standard deviation (n = 3).

## Data Availability

Not applicable.

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
