# Peer review of "Application of a Conducting Poly-Methionine/Gold Nanoparticles-Modified Sensor for the Electrochemical Detection of Paroxetine"

_polymers, 2021, doi:10.3390/polym13223981_

Round 1

Reviewer 1 Report

  1. The manuscript presented the modification of a GCE with AuNPs and DL-met to detect Paroxetine both isolated and in pharmaceutical formulations. The work presents interesting results, and the results are adequately described. 

    In my opinion, the present manuscript should be considered for publishing after some corrections; my comments are the following:

    1. Revise the English language in the entire manuscript. 
    2. Remove the description from the Materials and Methods section. 
    3. Table 1 must be placed in section 3.9
    4. The current description of GCE modification is insufficient. Authors should extend it, e.g., provide the CV of GCE in HAuCl4. Describe the deposition potential of Au in terms of overpotential (η). Also, provide the current transients of the Au electrodeposition.
    5. Why is there not a linear response between IPRX and Au deposition time? An analysis of AuNPs area should shed some light on this issue.
    6. AFM images must be improved because, with a wider window size than SEM, the surface features are not shown. 

Author Response

Thanks for your time and fruitful comments to enrich our manuscript, please find reply to comments and  amendments made in text.

  1. Revise the English language in the entire manuscript.

Done

  1. Remove the description from the Materials and Methods section.

Done

  1. Table 1 must be placed in section 3.7

Done

  1. The current description of GCE modification is insufficient. Authors should extend it, e.g., provide the CV of GCE in HAuCl4. Describe the deposition potential of Au in terms of overpotential (η). Also, provide the current transients of the Au electrodeposition.

Gold nanoparticles were deposited using chronoamperometry as mentioned in line 138 not using cyclic voltammetry. The i-t curve of deposition is currently given in the supplementary information. This approach for AuNPs deposition is adapted from previously reported methods as indicated in references 27, 29.

  1. Why is there not a linear response between IPRX and Au deposition time? An analysis of AuNPs area should shed some light on this issue.

In the fig. 1A IPRX was increased by 6-8 µA with increasing the Au deposition time from 10 - 30 s and then an insignificant decrease of~ 2 µA was noticed at 60 s of Au deposition. While 80 s deposition showed an increase of 12 µA after which the response decreased to 4 µA. The area of the modified surface of the electrode is shown in section 3.4 line 272 and 273 for the different stages of modification.

  1. AFM images must be improved because, with a wider window size than SEM, the surface features are not shown.

Done and the figures were updated

Reviewer 2 Report

In this work, designed and characterized Poly-Methionine/Au nanoparticles modified sensor for the electrochemical detection of Paroxetine. This work can be published after major revision. My comments and suggestions: 

1- In the introduction: please provide the main features of DL-methionine and why you used it with Au NPs for sensing application

2- Cite some of the many previously reported data about the use of Ag NPs for sensing applications and add statements for the novelty of your work

3- Please delate the lines from 95 to 110 (guidelines of the journal)

4- please provide the purity and company of all used chemicals and materials

5- Provide the EDX spectra as supplementary data if you will not include them in the manuscript.

6-Numbering of the figures inside the text is not consistent with the figure captions. I can not find Figure 4 in the text. So please carefully revise figures numbering in text and captions 

7- It is well known that the localized surface plasmon resonance  (LSPR) of Au NPs play important role in the sensing properties of Au NPs-based sensor, but in this manuscript, I could not see any discussions related to LSPR. Please highlight the role of LSPR and surface charge transfer in the sensing properties of your sensor. for example you can see the following papers for LSPR of Au nanostructures 

8- Please compare the sensing properties of your sensor with previously reported Au NPs-based sensors in the literature (Add a table)

9- Please discuss the reproducibility of your sensor

Author Response

Thanks for your time and fruitful comments to enrich our manuscript, please find replies to comments and amendments made in the text.

  • In the introduction: please provide the main features of DL-methionine and why you used it with Au NPs for sensing application
  • Done, and in the introduction page 2, in the 2nd, 3rd and 4th paragraphs, there is a short introduction on the main features of the application of electropolymerized amino acids including DL-methionine as efficient conducting polymers for different types of analytes

  • Cite some of the many previously reported data about the use of Au NPs for sensing applications and add statements for the novelty of your work

Examples are given in lines 86 and 87 and a novelty statement was given in lines 59-61.

  • Please delate the lines from 95 to 110 (guidelines of the journal)

           Done

4- please provide the purity and company of all used chemicals and materials

Done.

5- Provide the EDX spectra as supplementary data if you will not include them in the manuscript.

Added in the supplementary information file Fig.S-5

6-Numbering of the figures inside the text is not consistent with the figure captions. I can not find Figure 4 in the text. So please carefully revise figures numbering in text and captions

Done  

7- It is well known that the localized surface plasmon resonance  (LSPR) of Au NPs play important role in the sensing properties of Au NPs-based sensor, but in this manuscript, I could not see any discussions related to LSPR. Please highlight the role of LSPR and surface charge transfer in the sensing properties of your sensor. for example you can see the following papers for LSPR of Au nanostructures 

For sure localized surface plasmon resonance (LSPR) of Au NPs plays important role in the sensing properties of Au NPs-based optical sensors, but here we are dealing with an electrochemical rather than an optical sensor and its main role is to enhance the charge transfer which in turn enhances the sensitivity of the detection. That is why Au, is one of the metallic nanoparticles, commonly used in electrochemical sensors.

N.B.  no examples were given to refer to as indicated in the comment.

8- Please compare the sensing properties of your sensor with previously reported Au NPs-based sensors in the literature (Add a table).

Table 1 represents a comparison between the results of the present work and other previously reported electrochemical methods for PRX detection.  We think that it won't be that relevant to compare different analytes that used Au NPs!! especially the use of Au nanoparticles is not reported herein for the first time.

9- Please discuss the reproducibility of your sensor

Already discussed in page 12 second paragraph as highlighted in cyan.

Round 2

Reviewer 1 Report

Figure 3a(I) (i.e., AFM image of bare GCE) needs improvement because no surface features are seen in this image, and, although the GCE is pristine, with AFM, one can easily see the polishing lines and other surface features. This image does not show any of that.

Author Response

Done and the figure for AFM of GCE was replaced by a more clear one.

Reviewer 2 Report

Authors respond to all comments. I recommend the acceptance of this paper in present form

Author Response

Thank you for your acceptance recommendation